# Estimated Prevalence of Unreported IGD Cases in Routine Outpatient Children and Adolescent Psychotherapy

**DOI:** 10.3390/ijerph18136787

**Published:** 2021-06-24

**Authors:** Sonja Kewitz, Eva Vonderlin, Lutz Wartberg, Katajun Lindenberg

**Affiliations:** 1Institute of Psychology, Goethe University Frankfurt, 60323 Frankfurt am Main, Germany; kewitz@psych.uni-frankfurt.de; 2Centre for Psychological Psychotherapy Heidelberg, Heidelberg University, 69115 Heidelberg, Germany; eva.vonderlin@zpp.uni-hd.de; 3Department of Psychology, Faculty of Human Sciences, MSH Medical School Hamburg, 20457 Hamburg, Germany; lutz.wartberg@medicalschool-hamburg.de

**Keywords:** Internet Gaming Disorder, prevalence, children, adolescents, clinical diagnoses, clinical sample, comorbidities, F 63.8

## Abstract

Internet Gaming Disorder (IGD) has been included in the DSM-5 as a diagnosis for further study, and Gaming Disorder as a new diagnosis in the ICD-11. Nonetheless, little is known about the clinical prevalence of IGD in children and adolescents. Additionally, it is unclear if patients with IGD are already identified in routine psychotherapy, using the ICD-10 diagnosis F 63.8 (recommended classification of IGD in ICD-10). This study investigated N = 358 children and adolescents (self and parental rating) of an outpatient psychotherapy centre in Germany using the Video Game Dependency Scale. According to self-report 4.0% of the 11- to 17-year-old patients met criteria for a tentative IGD diagnosis and 14.0% according to the parental report. Of the 5- to 10-year-old patients, 4.1% were diagnosed with tentative IGD according to parental report. Patients meeting IGD criteria were most frequently diagnosed with hyperkinetic disorders, followed by anxiety disorders, F 63.8, conduct disorders, mood disorders and obsessive-compulsive disorders (descending order) as primary clinical diagnoses. Consequently, this study indicates that a significant amount of the clinical population presents IGD. Meaning, appropriate diagnostics should be included in routine psychological diagnostics in order to avoid “hidden” cases of IGD in the future.

## 1. Introduction

Gaming Disorder is the second behavioural addiction apart from Pathological Gambling included in the ICD-11 [1]. Additionally, Internet Gaming Disorder (IGD) can be found as a diagnosis for further study in the DSM-5 [2]. Due to its recent inclusion in diagnostic classification systems, prevalence is yet to be determined. Currently, broad ranges of prevalence estimations can be observed [3,4]. Fam [3] reports in his meta-analysis a pooled prevalence of 3.8% [confidence interval (CI) 95% = 2.5–5.2%] for children and adolescents between 10 and 19 years of age in studies since 2010—with an observed range of 0.6% to 15.7%. In total, estimations show that IGD is a relevant disorder in modern society, especially in youth [5]. Estimations in representative populations do not show the role IGD plays in in- and outpatient psychotherapy treatment populations.

To our knowledge clinical samples have not been considered in prevalence research on IGD so far. However, due to findings of high comorbidities in patients with IGD, a higher prevalence of IGD in clinical populations than in the general population is likely. Implications might be drawn from research on Internet Addiction which is a suggested umbrella syndrome of internet-related disorders, including IGD. Prevalence estimations of Internet Addiction in clinical samples of children and adolescents were reached via self-report [6,7,8] and one study used additional clinical interviews [9]. The prevalence estimations of Internet Addiction in clinical samples of children and adolescents are widely ranging from 11.2% to 24.1% [6,7,8,9]. This suggests that corresponding research on prevalence of IGD in clinical settings is necessary.

On top of that, this study targets children under the age of eleven since this age group has hardly been studied in the general population and even less in clinical samples. Paulus, Sinzig, Mayer, Weber, and Gontard [10] targeted a sample of four- to eight-year-olds in a general population-based sample. In Germany they found an estimated prevalence of IGD in 1271 children of 2.5% for boys and 1.4% for girls. All IGD criteria were assessed except the criterion: “Continued excessive use of Internet games despite knowledge of psychosocial problems” [2] (p. 795). Thus, they give a first insight into IGD in younger children. However, more research is needed to understand the onset of IGD and the degree of negative effect it has on young age groups. Through understanding the patterns of IGD, it will be possible to determine the best start for prevention and the necessity to adapt interventions for younger age groups. Therefore, this study contributes to this important gap in research.

Finally, this study aims to find out which clinical diagnoses patients meeting IGD criteria tend to be diagnosed with in current psychotherapeutic treatment. ICD-11, including the IGD diagnosis, will not be used as diagnostic classification system until 2022 [11]. Therefore, the diagnosis IGD cannot be given in current psychotherapeutic settings. Until ICD-11 is set in place, the diagnosis F 63.8 (“Other impulse disorders”) can be given to patients with IGD instead [12]. Therefore, a substitute diagnosis for IGD does exist. However, as stated previously, patients with IGD often get other primary clinical diagnoses causing the IGD to be “hidden”. Since IGD shows high comorbidities with multiple psychological disorders it is not clear which clinical diagnoses might frequently be given to patients with IGD. Reported comorbidities include depressive disorders, ADHD, obsessive-compulsive disorders, anxiety disorders, pathological gambling, and cannabis consumption [2,13,14,15,16,17,18,19,20,21]. Meaning, it is important to understand if patients with an IGD are already correctly identified so that they can receive specialised treatment. Otherwise regular diagnostics for IGD and education about IGD have to be set in place.

Since symptoms of IGD might affect progress in psychotherapy the aforementioned questions are important, yet to the best knowledge of the authors are unanswered by research. Consequently, the study at hand aims to answer the following research questions:How many 11- to 17-year-old patients that are being treated in an outpatient centre in Germany are affected by IGD?How many five- to 10-year-old patients that are being treated in an outpatient centre in Germany are affected by IGD?Which primary clinical diagnoses are patients affected by IGD diagnosed with?

## 2. Materials and Methods

### 2.1. Participants

The study was planned as a complete sample of children and adolescents between five and 17 years of age who were current patients (between March and November 2019) at the Center for Psychological Psychotherapy, Heidelberg University, Germany (ZPP; *Zentrum für psychologische Psychotherapie*). The ZPP is an outpatient centre where psychotherapists in training perform psychotherapy as part of their training. Since the ZPP also offers group therapy for Gaming and Internet Disorder the therapists are especially aware of IGD and its symptoms.

In total N = 526 patients were registered at the ZPP, ranging from age five to 17 (M = 11.33, SD = 3.55). The complete sample consists of 62.5% (*n* = 329) boys and 37.5% (*n* = 197) girls. Data concerning IGD were collected from N = 358 patients (*n* = 152 in the child and *n* = 207 in the adolescent sample with one patient being included in both samples because she turned 11 between parental and self-report), which is equivalent to 68.1% of the complete sample.

The analyses were divided into two groups, depending on age. Data for the five- to 10-year-olds (children) was collected through parental report. Of the total of N = 235 patients in that age group, data for 64.7% (*n* = 152) are available. At the time of the study, N = 291 11- to 17-year-old patients (adolescents) received treatment at the ZPP, of whom 71.1% (*n* = 207) filled in questionnaires concerning IGD (Video Game Dependency Scale; German abbreviation: CSAS). Their data were collected via youth’s self-report (CSAS-SR; *n* = 188) and via parental report (CSAS-PR; *n* = 188). A more detailed participant flow can be seen in Figure 1.

Since IGD data are not available for the complete sample, statistical analyses were conducted to search for differences between the complete and the incomplete sample. Results of patients with complete data differ significantly from patients with incomplete or no data on IGD concerning gender (CSAS-SR: Χ^2^ (1) = 5.31, *p* = 0.021; CSAS-PR adolescents: Χ^2^ (1) = 11.44, *p* = 0.001; CSAS-PR children: Χ^2^ (1) = 3.86, *p* = 0.050).

### 2.2. Measures

#### 2.2.1. Internet Gaming Disorder

IGD was assessed via the German version of the Video Game Dependency Scale (CSAS; Computerspielabhängigkeitsskala) [22]. The questionnaire is filled in by the adolescents themselves through self-report (CSAS-SR; CSAS-J) and by their parents through parental report (CSAS-PR; CSAS-FE). Due to cognitive competencies the questionnaire should explicitly not be filled in by children under the age of 10. The CSAS examines online and offline gaming behaviour. It takes about five to 10 minutes to complete the questionnaire. Both versions of the CSAS consist of 18 items, respectively two for every criterion of the DSM-5 diagnosis IGD. The items of the CSAS are answered on a four-point Likert scale ranging from “strongly disagree” to “strongly agree”, whereas according to the manual only the latter counts as a fulfilled criterion. A tentative diagnosis of IGD can—according to the manual—be given if at least five of nine criteria are met. When two to four criteria are met a patient is judged to be at risk of developing an IGD, according to the manual. The internal consistency for the CSAS-SR lies between α = 0.92 and α = 0.95, while there are no validation values for the CSAS-PR. [22]. The CSAS-SR has only been validated in a large-scale non-clinical adolescent sample. However, both versions of the CSAS show high reliabilities within the current clinical sample for both age groups (11- to 17- year old patients: CSAS-SR α = 0.94, CSAS-PR α = 0.97; 5- to 10-year-old patients: CSAS-PR α = 0.94). We further observed indications for a unidimensional factor structure for all three variations of the applied instrument (CSAS-SR: CFI: 0.93, TLI: 0.97 and RMSEA: 0.13/CSAS-PR for 11- to 17- year old patients: CFI: 0.99, TLI: 1.00 and RMSEA: 0.06/CSAS-PR for 5- to 10-year old patients: CFI: 0.96, TLI: 0.98 and RMSEA: 0.14).

Self-report data were filled in by adolescents between age 11 and 17. Parental report data were collected for all children and adolescents via the CSAS-PR. Even though the CSAS-PR is not designed for children under the age of 10, data in that younger age group were exploratively collected. This was done in order to generate data in this little considered group and, as mentioned above, a high reliability was reached in all age groups.

A measure for DSM-5 criteria was used due to other prevalence research being based on DSM-5 criteria and not ICD-11 criteria.

#### 2.2.2. Sociodemographic Data

Sociodemographic data were collected through basic documentation (BaDo) for each patient, including age, gender, and primary clinical diagnosis.

### 2.3. Procedure

The study at hand is a questionnaire-based study. All data were collected via paper-pencil questionnaires which were filled in by the children and adolescents and/or by their primary care givers. The data on IGD were collected between April 2018 and November 2019 from patients who were current patients at the ZPP between March and November 2019. Some patients had already filled in the CSAS prior to data collection. Their data were not re-collected but their original data were used. Age, gender and primary clinical diagnosis were retrieved from the basic documentation system. The IGD questionnaires were mostly filled in at home and partly during therapy sessions. Data assessed via the CSAS were used to indicate a tentative diagnosis of IGD.

### 2.4. Statistical Analysis

The estimated prevalence of IGD was calculated as percentage of patients meeting criteria for a tentative IGD diagnosis (5–9 criteria), at risk gaming behaviour (2–4 criteria) and normal gaming behaviour (0–1 criteria). This classification was used according to the CSAS manual [22]. For each category 95% confidence intervals were estimated. Primary clinical diagnoses of patients with a tentative IGD diagnose were analysed descriptively since only a few patients were fulfilling a minimum of five IGD criteria.

Cases with more than one missing value on the CSAS were excluded from analyses. In case of one missing value on the CSAS the missing value was—according to the manual—replaced by the patient’s mean on the remaining items of the CSAS [22]. Data from patients who filled in the CSAS completely, partially or not at all were compared concerning age and gender to analyse the generalisability of the sample at hand. The comparison concerning gender was conducted via an Χ^2^ test. Meanwhile, the comparison concerning age was conducted via *t*-tests between patients with complete versus partial and no data on the CSAS. These results are reported in section “Participants”.

## 3. Results

### 3.1. Estimated Prevalence of Internet Gaming Disorder (IGD)

As shown in Table 1, *n* = 7 of *n* = 177 adolescent patients met at least five diagnostic criteria of IGD according to CSAS-SR and can therefore be tentatively diagnosed with IGD. This corresponds to an estimated prevalence of 4.0% [CI 95% = 1.1–6.8]. In contrast, the estimated prevalence is 14.0% [CI 95% = 8.8–19.1] according to CSAS-PR as displayed in Table 2. For the children the estimated prevalence of IGD according to CSAS-PR lies at 4.1% [CI 95% = 0.9–7.4] as shown in Table 3.

Risky behaviour (2–4 criteria) is shown by 8.5% [CI 95% = 4.4–12.6] of the adolescent patients according to CSAS-SR. According to CSAS-PR the corresponding prevalence of at-risk patients is 12.8% [CI 95% = 7.8–17.8]. Finally, according to CSAS-PR 16.6% [CI 95% = 10.5–22.6] of the children show risky gaming behaviour.

To assess the congruence of self- and parental report the overlap of the two perspectives was observed. According to both, CSAS-SR and CSAS-PR *n* = 4 patients met 5 or more IGD criteria.

### 3.2. Primary Diagnoses of Patients Meeting at Least Five IGD Diagnostic Criteria

Table 4 shows the primary clinical diagnoses of patients meeting five or more IGD criteria. In the adolescent age group, 12.5% of all IGD cases (parental report), and 14.3% (self-report) were diagnosed with F63.8. Patients with a tentative diagnosis of IGD are mostly diagnosed with hyperkinetic disorders, followed by anxiety disorders, F 63.8, conduct disorders, mood disorders and obsessive-compulsive disorders (descending frequency).

## 4. Discussion

### 4.1. Estimated Prevalence in the Adolescent Patient Sample

This study shows that a relevant number of patients in a psychotherapy outpatient centre in Germany is affected by IGD. The estimated prevalence of IGD in adolescent patients aged 11 to 17 years is 4.0% according to self-reporting. According to parental report, the estimated prevalence in that age group goes up to 14.0%. The truth is likely to lie between these two perspectives, as parents tend to overestimate and adolescents to underestimate the severity of IGD symptoms. It is therefore reasonable that clinical judgment is usually reached through consideration of self- and third-party reports.

In order to understand the gap between perspectives it needs to be kept in mind that IGD is strongly associated with denial of symptoms [23]. Accordingly, Szász-Janocha, Vonderlin, and Lindenberg [24] found a prominent divergence between self- and parental report in patients at baseline. This inter-rater difference declined after a four-week intervention programme. Typically, IGD patients reported an increase of symptoms within the first four weeks of treatment, which was more likely explained by an increased illness insight rather than by actually worsened symptoms, because mid-term effects showed a strong symptom reduction. In contrast, parental ratings achieve much higher scores before treatment and a strong symptom reduction within the first weeks of treatment. Thus, in the study at hand self-report is likely to be an underestimation of IGD diagnostic criteria while parental report is likely to be an overestimation.

The current finding of 4.0% IGD prevalence in adolescents according to self-report is lower than estimated prevalence between 11.2% and 24.1% for Internet Addiction in clinical samples [6,7,8,9]. However, it needs to be kept in mind that IGD represents only a subgroup of the suggested umbrella syndrome Internet Addiction which includes other internet-related disorders. German large-scale studies reported a prevalence estimation of Internet Addiction in children and adolescents of 6.1% [25], opposed to 1.2% for IGD [26]. On top of that, the CSAS is a conservative measure of IGD that tends to reach lower prevalence estimations of IGD in representative samples than other measures do [26,27,28,29].

The best comparison between prevalence estimates in clinical and non-clinical samples can be drawn looking at studies that use identical measures: Rehbein, Kliem et al. [26] also used the CSAS via self-report as a measure of IGD and used the same cut-off scores for a diagnosis as this study did. In 11,003 13- to 18-year-old students the authors estimated the prevalence of IGD to be 1.2%. Comparing the current results to that estimation according to self-report, IGD would be more than three times more likely in a clinical sample than in a general population-based sample. However, the different sample sizes between the study of Rehbein, Kliem et al. [26] and the present study have to be kept in mind since smaller sample sizes tend to estimate higher prevalence of IGD than large sample sizes [3] which might lead to an overestimation of IGD in the current sample. In conclusion, this study suggests an IGD prevalence in a clinical population which is probably three times higher than in non-clinical populations and which lies between 4% (self-report) and 14% (parental report). Thus, it can be reasoned that IGD is a disorder that affects outpatient routine care to a significant extent and should be regularly considered in psychological diagnostics.

### 4.2. Estimated Prevalence in the Child Patient Sample

In the age group of the five- to 10-year-old patients, 4.1% are affected by IGD, according to parental report. This estimation is 3.4 times lower than the prevalence in the adolescent patient sample (14.0%) according to parental report. The result can also be compared to a general population-based study that estimated an IGD prevalence of 2.5% for boys and 1.4% for girls [10]. Thus, the estimated prevalence found in this clinical sample can be judged as higher than in the general population-based sample. Consequently, this study shows the relevance of IGD in a young age group and suggests that children within the clinical population are more likely to be affected by IGD than the general population. However, caution concerning generalisation is necessary: A higher percentage of boys was included in the investigated sample than in the complete sample which might lead to an overestimation of prevalence. Additionally, the CSAS is not designed for children under the age of 10. Thus, future research on IGD should figure out which criteria apply in that age group. For example, “Unsuccessful attempts to control the participation in Internet games” [2] (p. 795) might not be a sign of addiction but rather of not yet developed control competencies in a young age group. On top of that, parental report as mentioned above is likely to overestimate symptoms of IGD. Consequently, clinical interviews would be necessary to support tentative IGD diagnoses, especially in that age group.

### 4.3. Estimated Number of Hidden Diagnoses

Only 12.5% of all IGD cases (parental report) and 14.3% (self-report) were diagnosed with F 63.8 in the adolescent patient sample. In the child patient sample, none of the six patients were diagnosed with F 63.8. Even though we could not reach a clinical diagnosis due to missing expert rating, the data suggest an estimated number of undetected IGD cases between 85.7% and 100%.

It could be observed that more children and adolescents with a tentative IGD diagnosis were clinically diagnosed with hyperkinetic disorders and anxiety disorders than with F 63.8—the latter being used as a substitute diagnosis for IGD until ICD-11 is set in place [12]. Furthermore, conduct disorders, mood disorders and obsessive-compulsive disorders are observed as primary clinical diagnoses in patients showing at least five criteria of IGD. However, our findings contradict previous findings on comorbid IGD and depressive disorders [14,17,21]. In this study no patient with mood disorders met enough criteria for a full IGD according to self-report, only according to parental report. Even though the larger part of patients showing at least five IGD criteria was “hidden” behind other diagnoses, it was also observed that the diagnosis F 63.8 is being used correctly, according to the questionnaire-based data. The fact that most patients with a tentative IGD are diagnosed with another primary clinical diagnosis might align with reality since the given primary diagnoses could be more relevant than IGD in a particular patient. However, also secondary diagnoses were considered and only one patient had F 63.8 as a secondary diagnosis. Thus, it is likely that IGD in patients is in fact overlooked and thus considered too little in psychotherapy pointing out the importance of including IGD in routine diagnostics.

### 4.4. Limitations

Finally, it needs to be pointed out that the study is prone to limitations. Firstly, the CSAS was validated in a large-scale adolescent non-clinical sample, thus it has not been validated in a clinical sample. On top of that, the parental version has not been validated and is not designed for children under the age of 10 [22]. Nonetheless Cronbach’s alpha was very high in the current study for all age groups leading to the conclusion that the current data contribute to knowledge growth in the field. Also, in our sample, we observed indications for a unidimensional factor structure for all three variations of the CSAS, but the indices for the global goodness-of-fit of the model were not uniformly convincing. Therefore, further research is needed on the factorial structure of the instrument. On top of that, although the CSAS aims to cover all nine IGD criteria, current literature [30] recommends for example the Internet Gaming Disorder Scale (IGDS) [31] as a first-choice tool to assess all IGD criteria and all four GD criteria. Additionally, the CSAS can be considered a rather conservative measure of IGD [26], especially compared to the IGDS [27]. Therefore, the actual prevalence might be higher than reported in this study. More importantly, only tentative IGD diagnoses could be assessed since no diagnostic interviews were conducted. Therefore, we cannot interpret the IGD data as robust diagnoses reached through expert evaluation which leaves room for incorrect diagnoses. Thus, in future research it would be interesting to conduct clinical interviews to achieve valid diagnoses in order to replicate the current findings. Finally, some limitations concerning generalisation were found since gender differs significantly between the complete and the investigated samples.

## 5. Conclusions

Results show a relevant part of patients in a psychotherapy outpatient centre in Germany being affected by IGD. Adolescent patients between 11 and 17 years of age are affected 3.4 times more often than children aged five to 10 years. Patients with a tentative IGD diagnosis are most likely clinically diagnosed with other disorders than F63.8, i.e., hyperkinetic disorders, anxiety disorders, conduct disorders, mood disorders, or obsessive-compulsive disorders. Between 0% and 14.3% of patients who meet at least five IGD criteria are diagnosed with the substitute diagnosis F 63.8. Therefore, it is highly important to include IGD diagnostic tools in routine psychological diagnostics in order to obtain a thorough impression of patients and their symptomology. This could avoid unreported cases of IGD in the future.

## Figures and Tables

**Figure 1 ijerph-18-06787-f001:**
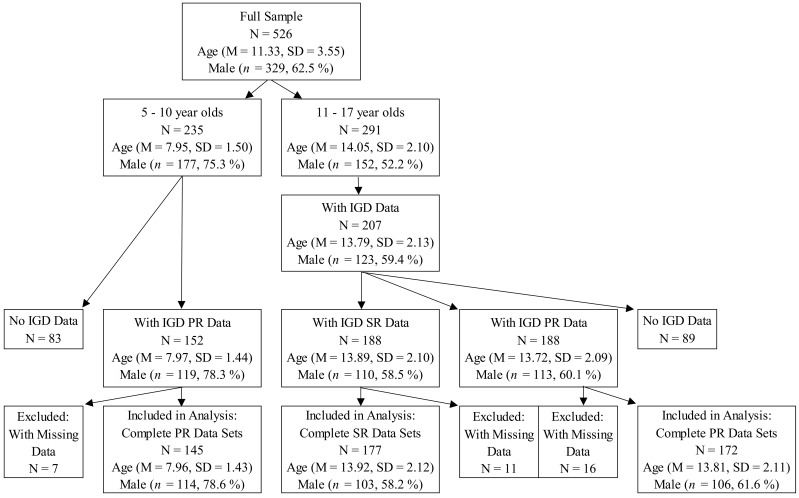
Participant flow of patients between five and 17 years of age at the ZPP (*Zentrum für psychologische Psychotherapie*) between March and November 2019. IGD: Internet Gaming Disorder. SR: self-report. PR: parental report.

**Table 1 ijerph-18-06787-t001:** Estimated Internet Gaming Disorder (IGD) prevalence in 11- to 17-year-old patients in outpatient psychotherapy assessed via self-report.

	Total Sample		Boys		Girls	
IGD Criteria	*N*	Prevalence Estimation	*N*	Prevalence Estimation	*N*	Prevalence Estimation
>5	7/177	4.0% [1.1, 6.8]	7/103	6.8% [1.9, 11.7]	0/74	0%
2–4	15/177	8.5% [4.4, 12.6]	14/103	13.6% [7.0, 20.2]	1/74	1.4% [−1.3, 4.1]
0–1	155/177	87.6% [82.7, 92.4]	82/103	79.6% [71.8, 87.4]	73/74	98.6% [96.0, 101.3]

**Table 2 ijerph-18-06787-t002:** Estimated IGD prevalence in 11- to 17-year-old patients in outpatient psychotherapy assessed via parental report.

	Total Sample		Boys		Girls	
IGD Criteria	*N*	Prevalence Estimation	*N*	Prevalence Estimation	*N*	Prevalence Estimation
>5	24/172	14.0% [8.8, 19.1]	22/106	20.8% [13.0, 28.5]	2/66	3.0% [−1.1, 7.2]
2–4	22/172	12.8% [7.8, 17.8]	19/106	17.9% [10.6, 25.2]	3/66	4.5% [−0.5, 9.6]
0–1	126/172	73.3% [66.7, 79.9]	65/106	61.3% [52.1, 70.6]	61/66	92.4% [86.0, 98.8]

**Table 3 ijerph-18-06787-t003:** Estimated IGD prevalence in 5- to 10-year-old patients in outpatient psychotherapy assessed via parental report.

	Total Sample		Boys		Girls	
IGD Criteria	*N*	Prevalence Estimation	*N*	Prevalence Estimation	*N*	Prevalence Estimation
>5	6/145	4.1% [0.9, 7.4]	6/114	5.3% [1.2, 9.4]	0/31	0%
2–4	24/145	16.6% [10.5, 22.6]	21/114	18.4% [11.3, 25.5]	3/31	9.7% [−0.7, 20.1]
0–1	115/145	79.3% [72.7, 85.9]	87/114	76.3% [68.5, 84.1]	28/31	90.3% [79.1, 100.7]

**Table 4 ijerph-18-06787-t004:** Primary clinical diagnoses of patients meeting >5 IGD criteria.

	Adolescents ^1^		Children ^2^
	SR ^3^	PR ^4^	PR^4^
F 63.8	1	3	0
Mood disorders ^5^	0	2	0
Anxiety disorders ^6^	2	6	2
Obsessive-Compulsive disorders ^7^	0	1	0
Hyperkinetic disorders ^8^	4	9	2
Conduct disorders ^9^	0	2	2
Other ^10^	0	1	0
Total	7	24	6

^1^ 11- to 17-year-old patients. ^2^ 5- to 10-year-old patients. ^3^ SR: Patients with a tentative IGD diagnose according to self-report. ^4^ PR: Patients with a tentative IGD diagnose according to parental report. ^5^ F 32.00, F 32.10. ^6^ F 40.10, F 41.10, F 43.20, F 93.20, F 93.80. ^7^ F42.10. ^8^ F 90.0, F 90.10. ^9^ F 91.0, F 91.3, F 92.0. ^10^ F 66.0.

## Data Availability

On request.

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
