# Peer review of "Estimated Prevalence of Unreported IGD Cases in Routine Outpatient Children and Adolescent Psychotherapy"

_ijerph, 2021, doi:10.3390/ijerph18136787_

Round 1

Reviewer 1 Report

Estimated Prevalence of Unreported IGD Cases in Routine Outpatient Children and Adolescent Psychotherapy

The topic of this manuscript is Internet Gaming Disorder (IGD) prevalence among youth in clinical population. Authors aimed (1) to assess prevalence of IGD diagnosis among 5-10 y.o. and 11-17 y.o. children treated in an outpatient setting and (2) highlight which primary clinical diagnoses had those children screened with IGD. Authors used Video Game Dependency Scale, filled by children themselves or/and parents (according to age groups), to screen for IGD. Study was done in a German setting for psychological psychotherapy. Overall, results showed a significant prevalence of IGD (from 4% to 14% according age group and self-/parents- evaluation). When comparing with patients’ files, IGD+ was screened with hyperkinetic disorders, followed by anxiety disorders, F63.8 (recommended classification of IGD in ICD-10), conduct disorders, mood disorders and obsessive-compulsive disorders (descending order) as primary clinical diagnoses.

The study delivers interesting data regarding IGD prevalence (a topic where such data are needed), among children in care for other psychological diagnosis. One of the most interesting results is that only 0% to 14.3% of those who meet at least five IGD criteria on screening scale had IGD diagnosis assessment in files, underlining the need of systematic assessment.

Here we suggest comments that should be addressed to authors:

Introduction: Authors refers to “internet addiction”, which is not an official diagnosis in DSM or ICD. e.g. “Implications might be drawn from research on Internet Addiction which is an umbrella syndrome of internet-related disorders, including IGD”. Authors should probably be more cautious here (like “proposed Internet Addiction”, or “hypothetical Interned addiction” formula, “suggested” instead of “is”, etc.)

Methods:

Figure 1 “participant flow” improve reading. Still, text indicates that “Data concerning IGD was collected from N = 358 patients”. Text should probably explain a little more, because how to calculate “358” is not clear when looking at Fig 1. (ex: 5-10 y.o : 235 – 83=152 had IGD data ; 11-17 y.o : 291-89=202 had IGD data ; however 152+202=354 (not 358)).

Measures: As the manuscript is in English, we suggest to use English abbreviations for readers’ immediate understanding and facilitate article search. So “CSAC”, (Computerspielabhängigkeitsskala), should probably be replaced by “VGDS” (Video Game Dependency Scale; Only mention that the German version was used and name it once in German).

Discussion:

In 4.3, Authors should emphasis that IGD+ in the study is a screening, not a diagnosis (notably because criteria were not confirmed by a trained clinician). In that perspective, this screening cannot strictly be the reference to label “incorrect” clinical diagnosis in patient’s files (the important result here is the absence of systematic assessment, more than if screening confirm or not IGD diagnosis). Comparison is interesting, but this could be more explained in Limits (“More importantly, only tentative IGD diagnoses could be assessed since no diagnostic interviews were conducted » à true fact, but consequence is missing).

Author Response

We thank the reviewer for this insightful feedback and hope that the changes are to their satisfaction.

Comment 1: Authors refers to “internet addiction”, which is not an official diagnosis in DSM or ICD. e.g. “Implications might be drawn from research on Internet Addiction which is an umbrella syndrome of internet-related disorders, including IGD”. Authors should probably be more cautious here (like “proposed Internet Addiction”, or “hypothetical Interned addiction” formula, “suggested” instead of “is”, etc.)

Response 1: We changed the sentence to the following: „Implications might be drawn from research on Internet Addiction which is a suggested umbrella syndrome of internet-related disorders, including IGD.” We therefore also changed the following sentence in the discussion: “However, it needs to be kept in mind that IGD represents only a subgroup of the suggested umbrella syndrome Internet Addiction which includes other internet-related disorders.” We hope that these changes clarify that Internet Addiction is not an official diagnosis.

Comment 2: Figure 1 “participant flow” improve reading.

Response 2: We changed the scale of Figure 1 and hope that it is now better readable.

Comment 3: Still, text indicates that “Data concerning IGD was collected from N = 358 patients”. Text should probably explain a little more, because how to calculate “358” is not clear when looking at Fig 1. (ex: 5-10 y.o : 235 – 83=152 had IGD data ; 11-17 y.o : 291-89=202 had IGD data ; however 152+202=354 (not 358)).

Response 3: We added an explanation as to how n = 358 is reached and also added and extra square in the participant flow for better understanding: „Data concerning IGD was collected from N = 358 patients (n = 152 in the child and n = 207 in the adolescent sample with one patient being included in both samples because she turned 11 between parental and self-report), which equivalates to 68.1 % of the complete sample.”

Comment 4: As the manuscript is in English, we suggest to use English abbreviations for readers’ immediate understanding and facilitate article search. So “CSAC”, (Computerspielabhängigkeitsskala), should probably be replaced by “VGDS” (Video Game Dependency Scale; Only mention that the German version was used and name it once in German).

Response 4: Thank you for your recommendation yet we would like to keep the abbreviation CSAS because it is also referenced this way in other English literature (see e.g. King et al., 2020: Screening and assessment tools for gaming disorder: A comprehensive systematic review).

Comment 5: In 4.3, Authors should emphasis that IGD+ in the study is a screening, not a diagnosis (notably because criteria were not confirmed by a trained clinician). In that perspective, this screening cannot strictly be the reference to label “incorrect” clinical diagnosis in patient’s files (the important result here is the absence of systematic assessment, more than if screening confirm or not IGD diagnosis).

Response 5: We changed the wording and do not speak of “correct” anymore except for the following sentence in which we added “according to questionnaire-based data”: “… it could also be observed that the diagnose F 63.8 is being used correctly, according to the questionnaire-based data.” Also, we added the following sentence: “Even though we could not reach a clinical diagnose due to missing expert rating, the data suggests an estimated number of undetected IGD cases between 85.7 % to 100 %.”       

Comment 6: Comparison is interesting, but this could be more explained in Limits (“More importantly, only tentative IGD diagnoses could be assessed since no diagnostic interviews were conducted » à true fact, but consequence is missing).

Response 6: We added the sentence: “Therefore we cannot interpret the IGD data as robust diagnoses reached through expert evaluation which leaves room for incorrect diagnoses. Thus, in future research it would be interesting to conduct clinical interviews in order to achieve valid diagnoses in order to replicate the current findings.”

Reviewer 2 Report

“The study was planned as a complete sample of children and adolescents between five and 17 years of age who were current patients (between March and November 2019)” and “The data on IGD was collected between April 2018 and November 2019.” The timeline of data collection is a little bit confusing. Please clarify.

It is not clear why the authors used the Video Game Dependency Scale which is based on DSM-5, instead of the WHO’s ICD-11 criteria. Please provide a better justification.

Did the authors mean that only endorsement of “strongly agree” counts as a fulfilled criterion. How about those who endorsed “agree”? Any references to support your this kind of classification?

“A tentative diagnose of IGD can be given if at least five of nine criteria are met. When two to four criteria are met a patient is judged to be at risk of developing an IGD.” please provide references for these two cutoff points.

It is strongly recommended to conduct EFA or CFA to identify the psychometric properties of the IGD scale since it has not been validated in the study population.

Author Response

We would like to thank the reviewer for his or her helpful comments and thorough review of our manuscript.

Comment 1: “The study was planned as a complete sample of children and adolescents between five and 17 years of age who were current patients (between March and November 2019)” and “The data on IGD was collected between April 2018 and November 2019.” The timeline of data collection is a little bit confusing. Please clarify.

Response 1: Thank you for pointing that ambiguity out to us. The different times have to do with how we collected the data. For better understanding we clarified: “The data on IGD was collected between April 2018 and November 2019 from patients who were current patients at the ZPP between March and November 2019. Some patients had already filled in the CSAS prior to data collection. Their data was not re-collected but their original data was used.”

Comment 2: It is not clear why the authors used the Video Game Dependency Scale which is based on DSM-5, instead of the WHO’s ICD-11 criteria. Please provide a better justification.

Response 2: For explanation we added the following sentence: “We used a measure for DSM-5 criteria instead of ICD‑11 criteria so that data could be better compared to other prevalence research since most research at the time of data collection had been done using DSM-5 criteria.”

Comment 3: Did the authors mean that only endorsement of “strongly agree” counts as a fulfilled criterion. How about those who endorsed “agree”? Any references to support your this kind of classification?

Response 3: This classification was based on the CSAS manual. For explanation we added “according to the manual”.

Comment 4: “A tentative diagnose of IGD can be given if at least five of nine criteria are met. When two to four criteria are met a patient is judged to be at risk of developing an IGD.” please provide references for these two cutoff points.

Response 4: This classification was based on the CSAS manual. For explanation we added “according to the manual”.

Comment 5: It is strongly recommended to conduct EFA or CFA to identify the psychometric properties of the IGD scale since it has not been validated in the study population.

Response 5: As recommended by Reviewer 2, we conducted CFA’s and included the results into the methods section. We observed indications for a unidimensional factor structure in the self-report version of the instrument (CSAS-SR) for the 11-to-17 year old patients (comparative fit index (CFI): 0.93, Tucker-Lewis index (TLI): 0.97 and a root mean square error of approximation (RMSEA): 0.13) and for the parental-report version (CSAS-PR) for the 11-to-17-year old patients (CFI: 0.99, TLI: 1.00 and RMSEA: 0.06). Our fit indices for the CSAS-SR (self-report) were comparable to those obtained in the original study by Rehbein and colleagues, who obtained a RMSEA of 0.02, a CFI of 0.99 and a TLI of 0.99 (Rehbein et al., 2015, p. 55).

However, for the parental-report-version for the 5-to-10 year old patients, we needed to exclude the item “My child is so frequently and intensively involved with computer and video games that he or she sometimes gets into trouble at school or work”, due do missing variance. The items of the CSAS are answered on a four-point Likert-scale ranging from “strongly disagree” to “strongly agree”, whereas according to the manual only the latter counts as a fulfilled criterion (recoded as 0=not fulfilling the criterion and 1 fulfilling criterion). Thus, no single 5-to-10 year old patient achieved a “strongly agree” answer (coded as 1) and thus, we needed to exclude this item from the factor analysis, which was then conducted with the 17 remaining items. Fit indices of the parental report (CSAS-PR) for 5- to 10-year old patients were CFI: 0.96, TLI: 0.98 and RMSEA: 0.14. We have included the CFA into the methods section and also discussed the limitations.

References

  1. Rehbein, F.; Baier, D.; Kleinmann, M.; Mößle, T. CSAS: Computerspielabhängigkeitsskala. Ein Verfahren zur Erfassung der Internet Gaming Disorder nach DSM-5. Hogrefe: Göttingen, Germany, 2015.

Round 2

Reviewer 1 Report

no further comments.

Author Response

Answer: Thank you for reviewing the manuscript.

Reviewer 2 Report

Thanks. I have no further comment.

Author Response

Answer: Thank you for reviewing the manuscript. We have checked English language and style by a native speaker.